# Recent decadal weakening of the summer Eurasian westerly jet attributable to anthropogenic aerosol emissions

Buwen Dong [1✉], Rowan T. Sutton [1], Len Shaffrey[1] & Ben Harvey [1]

The Eurasian subtropical westerly jet (ESWJ) is a major feature of the summertime atmospheric circulation in the Northern Hemisphere. Here, we demonstrate a robust weakening trend in the summer ESWJ over the last four decades, linked to significant impacts on extreme weather. Analysis of climate model simulations from the Coupled Model Intercomparison Project Phase 6 (CMIP6) suggests that anthropogenic aerosols were likely the primary driver of the weakening ESWJ. Warming over mid-high latitudes due to aerosol reductions in Europe, and cooling in the tropics and subtropics due to aerosol increases over South and East Asia acted to reduce the meridional temperature gradient at the surface and in the lower and middle troposphere, leading to reduced vertical shear of the zonal wind and a weaker ESWJ in the upper troposphere. If, as expected, Asian anthropogenic aerosol precursor emissions decline in future, our results imply a renewed strengthening of the summer ESWJ.

[1] National Centre for Atmospheric Science, Department of Meteorology, University of Reading, Berkshire, United Kingdom. ✉email: b.dong@reading.ac.uk

The Eurasian subtropical westerly jet (ESWJ), a narrow band of a strong wind blowing from west to east with large horizontal and vertical wind shear, exists in the upper troposphere and lower stratosphere over the Eurasian continent all year round with large seasonal variations in its latitude and strength[1–3]. The summer ESWJ is centred at ~40°N, features zonal wind speeds exceeding 25 m s$^{-1}$ with maxima over West and East Asia associated with the blocking effect of the Tibetan Plateau[4,5], and acts as a guide for Rossby waves[6,7]. Its formation and variability are closely related to the thermal contrast between the tropics and subtropics and therefore to the intensity and position of the meridional temperature gradient (MTG) in the troposphere[1–3]. Variations of the summer ESWJ have been demonstrated to affect mid-latitude weather and climate over Asia[8–13].

There is evidence that the zonal mean summer circulation and the East Asian subtropical westerly jet in the northern mid-latitudes have weakened in the past 40 years[14–16] and that this weakening may have led to important effects on weather, such as a reduced number of strong extratropical cyclones[17,18], increased prolonged summer weather extremes in mid-latitudes[19–21], and decreasing rainfall trends over Central Asia[22]. Several studies have attributed the weakening of northern summer circulation to amplified Arctic warming associated with the climate system's response to increases in greenhouse gases (GHG)[14,20,23]. The theoretical basis is that reduced MTG in the lower troposphere leads to reduced vertical shear of the zonal wind through thermal wind balance, and therefore weakens the upper tropospheric jet. However, GHG forcing also leads to enhanced warming in the tropical upper troposphere associated with enhanced convection and latent heat release[24]. This leads to an increase in MTG in the upper troposphere which may counter the influence of the lower-tropospheric MTG, establishing a potential "tug-of-war" governing the response of the upper tropospheric jet to increases in GHG[25,26]. Note that a focus on the zonal mean wind masks significant regional variations in summertime circulation change[15,16], which have yet to be explained.

Anthropogenic aerosols affect global and regional climate through aerosol-radiation and aerosol-cloud interactions[26]. Because of their inhomogeneous spatial distributions, aerosols can cause changes in horizontal and vertical temperature gradients[27–35], which in turn affect atmospheric circulation, potentially including the strength and position of subtropical jet streams in the Northern[27,30–34] and Southern[34,35] Hemispheres. During the past few decades, there were large changes in the magnitude and spatial patterns of aerosols and their precursor emissions. Emission changes were characterized by reductions over North America and Europe since the mid-1980s in response to air quality measures and increases over Asia and Africa[36]. It has been shown that this pattern of aerosol emissions influenced wintertime extreme weather events via its influence on the large-scale atmospheric circulation and mid-latitude dynamics[37].

In this study, we investigate evidence that the changes in aerosol emissions have exerted an influence on the northern hemisphere summertime atmospheric circulation, and in particular the ESWJ. We analyse trends in the summer ESWJ during the last four decades (1979/1980–2019) in four reanalysis data sets (ERA5, NCEP, JRA55, and MERRA2[38–41]) and identify a significant weakening trend. We then use multimodel simulations from the coupled model intercomparison project phase 6 (CMIP6)[42] to identify the causes of this weakening trend (see Methods).

## Results

**Weakening trend in the Eurasian subtropical westerly jet.** The mean structure of the summer ESWJ in the ERA5 reanalysis shows a maximum speed at approximately 200 hPa and 40°N (Fig. 1a, b). We define an ESWJ index by averaging the zonal wind speed over the region shown in Fig. 1a, which spans the local maxima over East and West Asia; this index has a mean value of 27.1 m s$^{-1}$. During 1979–2019 there were substantial changes in the upper tropospheric zonal winds which are robust in all four reanalyses (Fig. 1c–e and Supplementary Figs. 1, 2). These changes involve a significant weakening trend in the westerly jet with an equivalent barotropic vertical structure. The largest decreases are seen in the upper troposphere close to the core of the jet around 40°N, and on its equatorward side. The trend and the 90% confidence interval in the summer ESWJ index over 1979–2019 are $-0.43 \pm 0.36$ m s$^{-1}$ decade$^{-1}$ for ERA5 ($-0.45 \pm 0.35$, $-0.40 \pm 0.36$ and $-0.63 \pm 0.37$ m s$^{-1}$ decade$^{-1}$ for NCEP, JRA55 and MERRA2 during 1980–2019, respectively) with a mean trend of $-0.48$ m s$^{-1}$ decade$^{-1}$ with ERA5, NCEP and JRA5 showing similar magnitudes (see Methods), corresponding to a weakening of approximately 2.0 m s$^{-1}$ or about 7% of the mean jet-speed over the 41 year period. Accompanying the weakening of the Eurasian subtropical westerly jet is an enhancement of the East Asian subpolar jet at about 60°N, indicating concurrent variation of the subtropical and subpolar jets over East Asia[12,13,16].

**Attribution of the observed trend in the Eurasian subtropical westerly jet.** To investigate the causes of the observed weakening trend in the ESWJ we analyse eight multimodel CMIP6 historical simulations[42] for the period 1979–2014 with different external forcing factors (see Methods and Supplementary Table 1). The multimodel mean (MMM) climatological summer ESWJ index is 24.2 m s$^{-1}$ for historical simulations using ALL forcings, with similar climatological values (24.2, 23.6 and 23.6 m s$^{-1}$, respectively) for DAMIP single forcing (GHG, AER and NAT forcings) simulations[43]; these values are slightly weaker than was found in reanalyses (27.1 m s$^{-1}$). For the ALL simulations, MMM trends at 200 hPa show a weakening of the westerlies along the subtropical westerly jet core and an enhancement of westerlies northward at about 60°N (Fig. 2a). The sector-averaged zonal winds show the weakening is confined to the upper troposphere with the largest changes seen on the equatorward flank and just below the subtropical jet core (Fig. 2b). The similarity between this spatial structure seen in the model simulations and the structure seen in the reanalyses (Fig. 1) suggests that common factors may be responsible.

To identify the contribution of individual forcings, namely GHG, AER and NAT, to the changes in the ESWJ, we use DAMIP single forcing experiments[43]. MMM results indicate a weak impact of GHG changes at 200 hPa (Fig. 2c), with an indication of a strengthening of the westerlies on the equatorward side of the jet. The strongest changes are increases in the westerly wind speeds above the subtropical jet core in the upper troposphere and lower stratosphere (Fig. 2d). By contrast, the changes simulated in AER simulations (Fig. 2e, f) are in many respects very similar to those seen in the ALL simulations: both the horizontal structure at 200 hPa and the vertical structure in the sector average show good agreement. A small difference is that the strongest anomalies in AER simulations are located in the subtropical jet core, whereas in the ALL simulations the strongest anomalies are found slightly below and equatorward of the subtropical jet core. The responses to AER forcing in individual models all show weakening trends in the westerly winds and an equivalent barotropic structure with maximum values close to the subtropical jet core in the upper troposphere and weak enhancement of westerlies at about 60°N for most models, although there are some differences in the magnitude of the simulated trends (Supplementary Figs. 3, 4). The NAT-induced

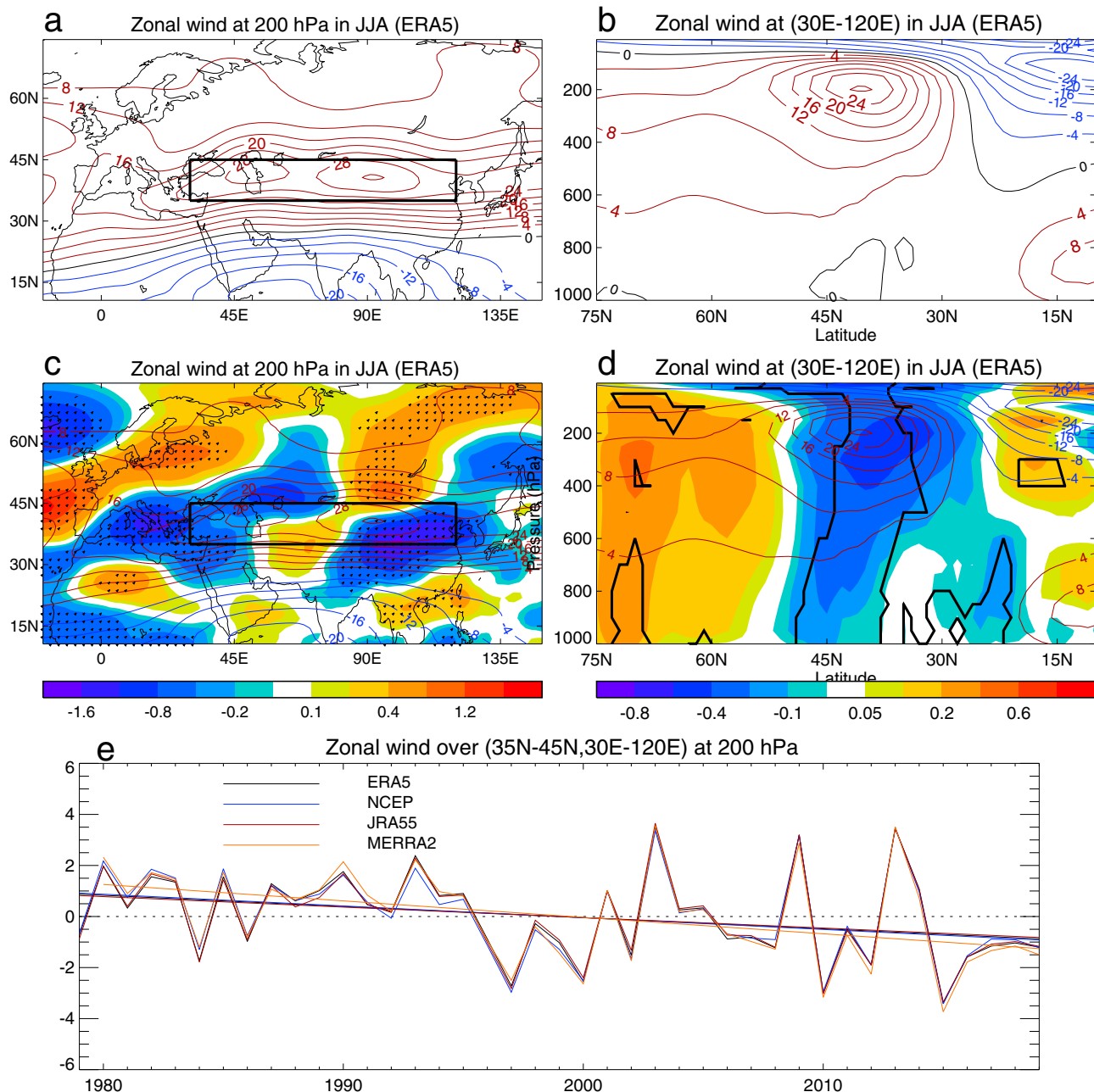

**Fig. 1 The linear trends of zonal wind in summer (June, July, August) during the last four decades. a**, **b** Climatology (m s$^{-1}$) (time mean over 1979–2019) at 200 hPa and at the latitude-height (hPa in pressure coordinate) cross-section zonally averaged over the Eurasian sector (30°E–120°E) based on ERA5 reanalysis. **c**, **d** Linear trends (m s$^{-1}$ decade$^{-1}$) of zonal wind at 200 hPa and at the latitude-height cross-section over the Eurasian sector with contours showing climatology. **e** Time series of the ESWJ index, defined as the area-averaged zonal wind over the region 35°N–45°N, 30°E–120°E (block box in **a**, **c**) at 200 hPa, based on four reanalyses and corresponding linear trends during 1979–2019 (1980–2019 for MERRA2). The linear trends of the ESWJ index are −0.40 to −0.63 m s$^{-1}$ decade$^{-1}$ with a mean of −0.48 m s$^{-1}$ decade$^{-1}$. Dots in **c** and thick black lines in **d** indicate regions where trends are statistically significant at the 10% level using the Mann–Kendall test. See Methods for details of data sets and analysis.

changes are small (Fig. 2g, h). These results strongly suggest that AER forcing changes are the primary driver of the observed weakening trend in the ESWJ over the recent decades (Fig. 2).

Quantitative comparisons between the trends in the summer ESWJ index in the four reanalyses and model simulations are shown in Fig. 3. Considering ensemble means for each model, the trend in the ALL simulations ranges from −0.63 to −0.07 m s$^{-1}$ decade$^{-1}$ (Fig. 3a). The MMM ESWJ index trend and its uncertainty exhibit a trend of −0.34 ± 0.07 m s$^{-1}$ decade$^{-1}$ (see Methods), which is slightly weaker (by about 30%) than the mean value based on the

four reanalysis data sets (−0.48 m s$^{-1}$ decade$^{-1}$). The GHG and AER forcings have opposite effects on the strength of the summer ESWJ index. AER forcing induces a weakening of the ESWJ index with a range of −0.46 to −0.11 m s$^{-1}$ decade$^{-1}$ with a MMM value of −0.36 ± 0.04 m s$^{-1}$ decade$^{-1}$, which is very similar to that found in the ALL simulations. In contrast, GHG forcing leads to a strengthening of the ESWJ index of 0.12 ± 0.04 m s$^{-1}$ decade$^{-1}$ in the MMM with a multimodel range of −0.01 to 0.26 m s$^{-1}$ decade$^{-1}$. NAT forcing leads to a weak response of 0.06 ± 0.06 m s$^{-1}$ decade$^{-1}$ with a range of −0.24 to 0.21 decade$^{-1}$.

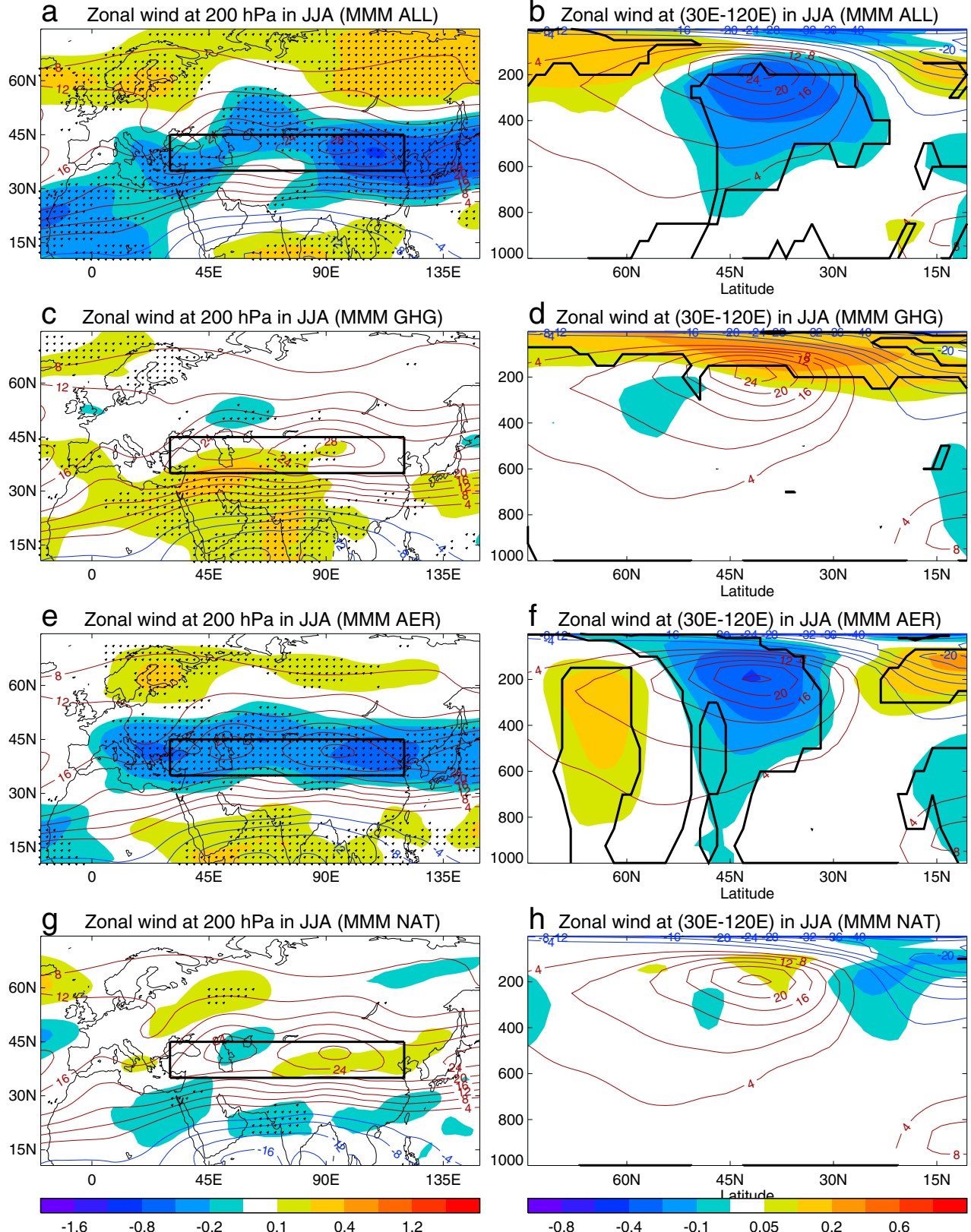

**Fig. 2 The multimodel mean (MMM) linear trends of zonal wind in summer (June, July, August) during 1979–2014 in CMIP6 (DAMIP) simulations.**
**a**, **c**, **e**, **g** Linear trends (m s⁻¹ decade⁻¹) at 200 hPa. **b**, **d**, **f**, **h** Linear trends (m s⁻¹ decade⁻¹) at the latitude-height (hPa in pressure coordinate) cross-section zonally averaged over the Eurasian sector (30°E–120°E). Contours show the corresponding climatology and black boxes (left) highlight the region (35°N–45°N, 30°E–120°E) where ESWJ index is calculated. **a**, **b** ALL simulations. **c**, **d** GHG simulations. **e**, **f** AER simulations. **g**, **h** NAT simulations. Dots (left) and thick black lines (right) indicate regions where at least seven out of eight models show the same sign of trends. Only seven models are included in panels **g** and **h** for NAT simulations with dots showing six out of seven models showing the same sign of trends. See Methods for details of model simulations and analysis.

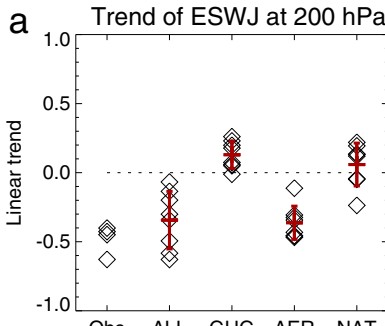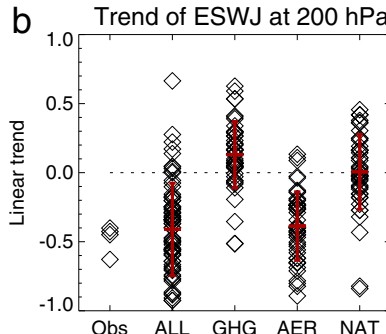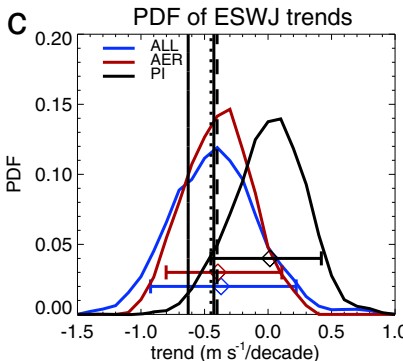

**Fig. 3 Linear trends of summer (June, July, August) ESWJ index in four reanalyses and in CMIP6 (DAMIP) simulations. a** Linear trends (m s⁻¹ decade⁻¹) in reanalyses and the ensemble means of each model, with diamonds indicating each reanalysis (Obs) and each of the eight models for different forcing simulations (ALL, GHG, AER, NAT). **b** Linear trends (m s⁻¹ decade⁻¹) in reanalyses and all model ensemble members. The pluses show the multimodel or multi-ensemble means and red lines indicate one standard deviation range. Trends in reanalyses are during 1979–2019 (1980–2019 for MERRA2) and in model simulations are during 1979–2014. **c** Kernel fitted probability density function (PDF) for linear trends of summer ESWJ index for ALL, AER simulations during 1979–2014, and using 36-year-long non-overlapping segments extracted from piControl (PI) simulations. The thick horizontal lines denote the 5th to 95th percentile range for the distribution with diamonds showing median. Four vertical lines are trends based on four reanalyses. See Methods for details of data sets and model simulations and analysis.

Results based on individual ensemble members for each model show similar behaviour although, as expected, the ranges of model-simulated trends for each force are larger than those based on model ensemble means (Fig. 3b). Figure 3c shows a comparison between the observed trend in the ESWJ index and trend distributions derived from the ALL and AER simulations and from pre-industrial (PI) control simulations (see Methods). The probability of observing a weakening trend as large as that in ERA5, NCEP and JRA55 reanalyses in an unperturbed (PI) climate is estimated to be 6.5%, whereas such trends are far more likely in the ALL and AER simulations. Added to the evidence from Figs. 1, 2 concerning the spatial structure of observed and simulated changes, these results support a conclusion that the observed weakening of the ESWJ was likely primarily driven by AER changes.

**Physical mechanisms.** During the past four decades, concentrations of greenhouse gases have continued to increase[42–44]. During the same period, there have been large changes in the magnitudes and spatial patterns of anthropogenic aerosol precursor emissions, characterised by decreases over Europe and North America and increases over South and East Asia and Africa[32,36,37]. The changes in aerosol emissions lead to changes in aerosol optical depth (AOD) characterised by decreases over North America and Europe and increase over East Africa, the Middle East, South Asia, and East Asia in both MERRA2 reanalysis and DAMIP AER simulations (Supplementary Fig. 5a, e). The impacts of changes in both anthropogenic and natural forcings on downward surface solar radiation (SSR), surface air temperature and precipitation in the CMIP6 simulations are shown in Fig. 4. The SSR trends feature a striking dipole pattern with positive trends over Europe

and northern Asia and negative trends over Africa, South Asia and East Asia in ALL simulations (Fig. 4a). This dipole pattern of SSR trends is predominantly attributed to changes in AER forcing with GHG and NAT-induced trends being very weak (Fig. 4a, d, g, j).

Observations and reanalyses show large warming trends over North Africa, southern Europe, the Middle East and East Asia with weak trends over part of North America and over Central Asia (Supplementary Fig. 5b–d). Responses to All forcing changes show similar warming trends over North Africa, southern Europe, the Middle East and East Asia, but warming trends over North America and Central Asia are overestimated (Fig. 4b and Supplementary Fig. 5f). GHG forcing leads to widespread warming of SAT, but the AER-induced trends in SSR lead to spatially inhomogeneous trends in SAT, characterised by enhanced warming over mid-high latitude Eurasia and reduced warming over tropical Africa, South and East Asia (Fig. 4e, h). Precipitation changes (Fig. 4c, f, i, l) in the ALL simulations show increases over Africa, South Asia and northern East Asia. The GHG and AER simulations imply that changes in both greenhouse gases and anthropogenic aerosols influence these changes, with aerosols being dominant over Africa and both forcing factors being important over Asia. The influence of NAT forcing on both SAT and precipitation is weak.

The inhomogeneous SAT trends in ALL and AER simulations (Fig. 4b, h) are associated with inhomogeneous temperature changes in the troposphere (Supplementary Figs. 6, 7) with enhanced warming over mid-high latitudes in the lower and mid-troposphere and relatively weak warming in the tropics over the Eurasian continent. These temperature trends imply a reduced meridional temperature gradient (MTG) throughout most of the troposphere with weakening trends peaking in mid-latitudes at

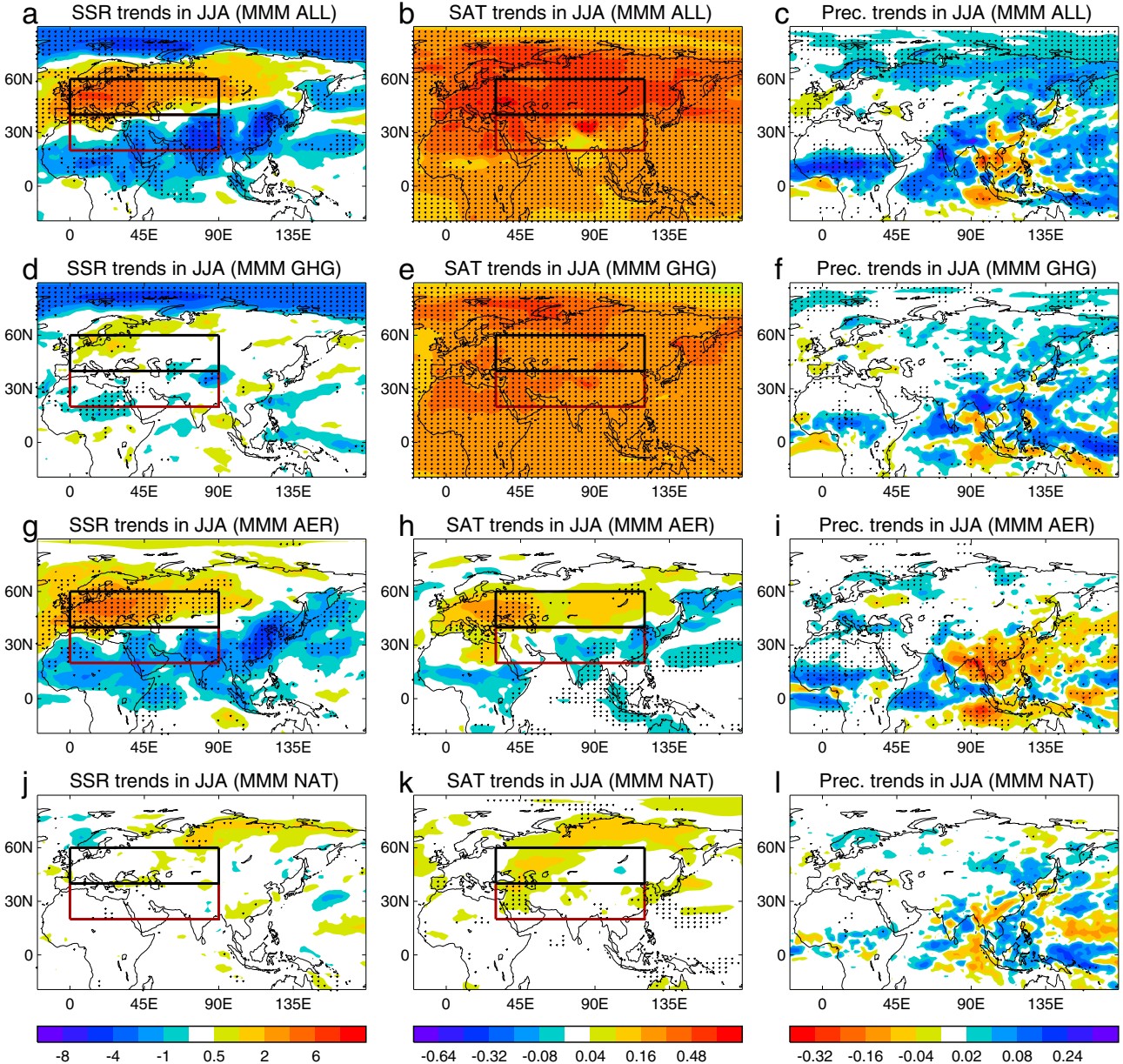

**Fig. 4 Spatial patterns of multimodel mean (MMM) linear trends of some surface variables in summer (June, July, August) for CMIP6 (DAMIP) simulations during 1979–2014. a, d, g, j** Downward surface solar radiation (SSR, W m$^{-2}$ decade$^{-1}$). **b, e, h, k** Surface air temperature (SAT, °C decade$^{-1}$). **c, f, i, l** Precipitation (mm day$^{-1}$ decade$^{-1}$). **a–c** ALL simulations. **d–f** GHG simulations. **g–i** AER simulations. **j–l** NAT simulations. Dots indicate regions where at least seven out of eight models have the same sign of trend. Red and black boxes (left panels) highlight regions of (20°–40°N, 0°–90°E) and (40°–60°N, 0°–90°E) that are located to the south and north of the summer Eurasian climatological jet core and used to define the SSR gradient index (south box minus north box). Red and black boxes (middle panels) highlight regions (20°–40°N, 30°–120°E) and (40°–60°N, 30°–120°E) that used to define SAT gradient index. Only seven models are included for NAT simulations with dots showing six out of seven models showing the same sign of trends (**j–l**). See Methods for details of model simulations and analysis.

30°−50°N (Fig. 5a, b, g, j). Through thermal wind balance, this reduced MTG is associated with reduced vertical shear of the zonal wind and hence a weakening of the ESWJ in the upper troposphere (Figs. 5c, i, 2b, f).

In response to changes of GHG forcing, similar to SAT trends, temperature trends in the lower and mid-troposphere are relatively uniform over the Eurasian continent (Supplementary Figs. 6, 7). As a result, trends of MTG in the lower and mid-troposphere are weak, as are trends in zonal winds (Figs. 5e, f, 2d). In the upper troposphere GHG forcing leads to an increased MTG (Fig. 5e)

associated with enhanced warming in the tropics associated with convection and latent heat release[24,25] (Supplementary Fig. 6f and Fig. 7b). This increased MTG is consistent with the enhanced zonal wind in the upper troposphere and lower stratosphere (Fig. 5f).

Despite the fact that the temperature changes in the ALL simulations are superficially most similar to those in the GHG simulations (Supplementary Figs. 6, 7), changes in the tropospheric zonal wind, including the ESWJ, are much more similar to those in the AER simulations. The reason is that the relevant wind changes are governed by changes in the tropospheric MTG,

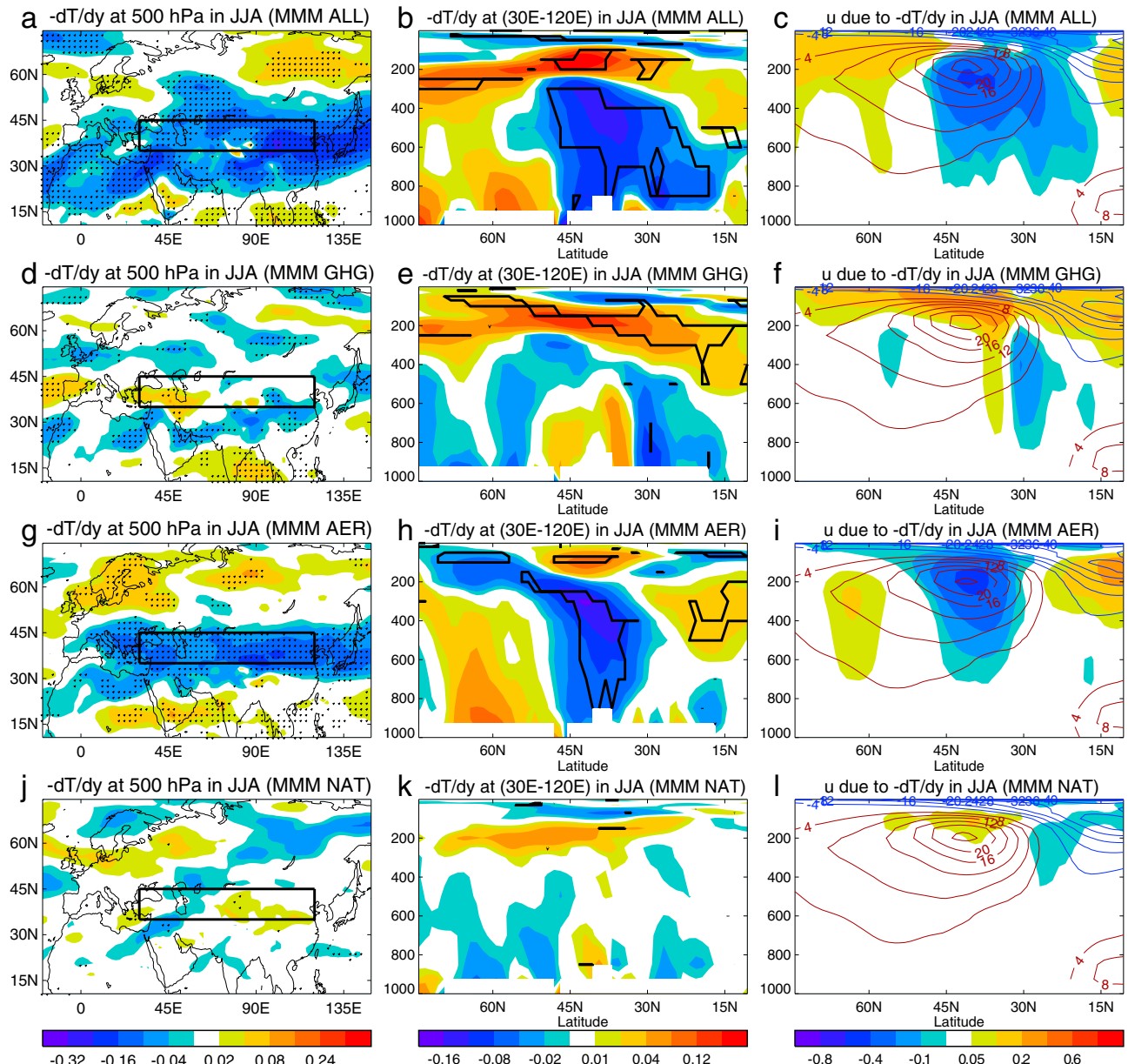

**Fig. 5 The multimodel mean (MMM) linear trends of meridional temperature gradient (MTG) in summer (June, July, August) during 1979–2014 in CMIP6 (DAMIP) simulations. a, d, g, j** Linear trends (K per 1000 km decade$^{-1}$) at 500 hPa. **b, e, h, k** Linear trends (K per 1000 km decade$^{-1}$) at the latitude-height (hPa in pressure coordinate) cross-section zonally averaged over the Eurasian sector (30°E–120°E). Note shown in these panels are −dT/dy which is positive for the climatology in mid-high latitudes over the Northern Hemisphere. **c, f, i, l** are trends (m s$^{-1}$ decade$^{-1}$) of zonal winds derived based on the thermal wind balance from the cross-section of MTG in middle panels (**b, e, h, k**). **a–c** ALL simulations. **d–f** GHG simulations. **g–i** AER simulations. **j–l** NAT simulations. Dots (left panels) and thick black lines (middle panels) highlight regions where at least seven out of eight models show the same sign of trends. Only seven models are included in panels **b, c, j–l**. Dots in **j** and thick black lines in **b, k** highlight regions where at least six out of seven models show the same sign of trends. See Methods for details of model simulations and analysis.

which are much more pronounced in AER than in GHG simulations. A further implication of these results is that changes in tropical precipitation, which exhibit some differences between the ALL and AER simulations (Fig. 4c, i), do not play a major role in the weakening of the ESWJ.

To further assess the importance of the MTG we calculated the residual differences between the linear trends of zonal wind (right column of Fig. 2) and zonal wind trends derived from the trends of MTG based on the thermal wind balance (right column of

Fig. 5). In all cases, the magnitude of the residual trends is small (Supplementary Fig. 8) in comparison with the thermal wind contribution. These results indicate that the trends in the ESJW are primarily determined by MTG changes in the lower and middle troposphere[2].

The inter-model variation between AER simulations is also consistent with the mechanism for ESWJ change identified in the MMM response (Supplementary Fig. 9). Considering ensemble means for each model, the correlation between the weakening

trend in the ESWJ index and the trend in the meridional gradient in SAT is 0.93. There is also a very high correlation (0.97) between the northern hemisphere aerosol effective radiative forcing (ERF) in summer for each model and the corresponding meridional gradient in SSR. The correlation between SAT and SSR gradients is lower (0.57), which suggests that other inter-model differences, e.g. in the representation of the land surface[45], also contribute to the inter-model variation in SAT gradient and hence ESWJ.

**Relationship between trends in the Eurasian jet and trends in zonal mean zonal winds**. As discussed earlier, previous studies have highlighted a weakening of the summertime zonal mean circulation since 1979 and suggested that enhanced warming of the Arctic in response to increases in greenhouse gas concentrations could be responsible[14,20]. Supplementary Fig. 10 shows the trends in the zonal mean zonal winds and a decomposition into Eurasian, non-Eurasian, western and eastern hemisphere sectors for the period 1979–2019. The zonal mean shows significant weakening on the equatorward side of the jet in the upper troposphere, which reflects the changes over the Eurasian and eastern hemisphere sectors. There is an additional weakening feature on the poleward side of the subtropical jet at around 60°N, which is not significant in the zonal mean, but is significant in a mean over the non-Eurasian and western hemisphere sectors. Closer inspection shows that this feature is primarily a reflection of an equatorward shift of the jet over the North Atlantic sector[33]. Coumou et al.[14] focused on changes in zonal mean zonal wind at 500 hPa averaged over the latitude band 35°–70°N. Supplementary Fig. 10 shows that such an average combines contributions from two separate features related to the western and eastern hemisphere sectors with changes over the Eurasian sector showing similar features as the eastern hemisphere sector.

Supplementary Fig. 11 shows that the CMIP6 MMM reproduces a weakening on the equatorward side of the zonal mean jet in the upper troposphere and that this feature is attributable to response over the Eurasian sector to AER forcing, as previously discussed. The equatorward shift of the jet over the Atlantic Sector is not reproduced in the CMIP6 MMM. This might be because it reflects internal variability in the real world or a forced response that is not captured by the CMIP6 models[33]. However, the CMIP6 results do not support the suggestion of Coumou et al.[14] that weakening of the summertime zonal mean circulation is attributable to rapid warming in the Arctic in response to GHG forcing. Rather, they suggest that AER forcing was more important than GHG forcing for explaining changes in the zonal mean summer circulation in recent decades.

## Discussion

Our results have demonstrated a robust weakening of the summer ESWJ over the last four decades from 1979 to 2019. CMIP6 simulations provide compelling evidence that changes in anthropogenic aerosol precursor emissions were the primary driver of this weakening. Therefore, this change is a striking example of how human activities can affect atmospheric circulation on a continental scale. It is particularly notable that the ESWJ appears to have been much more strongly influenced by anthropogenic aerosols than by the increasing concentrations of greenhouse gases. We have also shown that changes in the ESWJ and its zonal extension make an important contribution to changes in the zonal mean zonal winds over recent decades. Our evidence for the importance of anthropogenic aerosols contrasts with suggestions that weakening of the zonal mean winds might have been caused by rapid warming of the Arctic[14,20,23].

It is highly likely that the weakening of the ESWJ has had significant impacts on regional weather: in particular, it may have led to a reduced number of strong extratropical cyclones[17,18], increased prolonged summer weather extremes in mid-latitude[19–21], and decreasing rainfall trends over Central Asia[22]. There is a need for more research to understand the dynamic relationships between the ESJW and regional weather. Over the next few decades, it is expected that Asian anthropogenic aerosol precursor emissions will decline, while greenhouse gas concentrations continue to increase. Our results imply that the ESWJ is likely to strengthen again over this time period, potentially reversing some of its impacts on regional weather.

Our finding that anthropogenic aerosols are the primary driver of the weakening of the summer ESWJ over the last four decades is based on the CMIP6 multimodel ensembles. There is some evidence that CMIP6 models may overestimate the 1960–1990s cooling over the northern hemisphere mid-latitudes during the mid-20th century related to stronger aerosol effective radiative forcing[46,47] in comparison with CMIP5 models[48,49] and/or higher climate sensitivity[46,49]. CMIP6 simulations may also underestimate the recent (2006-2014) decrease in aerosol emissions over East Asia[50]. These factors might mean that the aerosol-forced ESWJ weakening might be somewhat stronger in the CMIP6 models than in the real world. However, in view of the distinctive fingerprint of AER forcing identified in our research, it is unlikely that improved models would modify our conclusion that anthropogenic aerosol precursor emissions were the primary driver of the weakening of the summer ESWJ over the last four decades.

## Methods

**Reanalysis and observation data sets**. The reanalysis data sets used in this study are monthly mean zonal winds on pressure levels and surface air temperature (SAT) from the new state-of-the-art climate reanalysis of the European Centre for Medium-Range Weather Forecast (ERA5)[38], the National Center for Environmental Prediction (NCEP Reanalysis)[39], the Japanese 55-year Reanalysis Project (JRA55)[40] during 1979–2019, and the Modern-Era Retrospective Analysis for Research and Applications, version 2 (MERRA2)[41] during 1980–2019. The month mean aerosol optical depth (AOD) from the MERRA2 is also used. The observed monthly mean SAT is from the CRU TS4.05 data set during 1979–2019[51]. These reanalysis and observation data sets were interpolated to a common grid with a horizontal resolution of 1.875° longitude by 1.25° latitude. We used monthly mean data to construct summer (June, July, August) means and investigated the trends of westerlies by analysing linear trends of zonal winds in summer at 200 hPa over Eurasia in the Northern Hemisphere, sectorially averaged zonal winds over the Eurasian sector (30°E–120°E) and the summer Eurasian subtropical westerly jet (ESWJ) index, defined as the area-averaged zonal wind over the Eurasian climatological jet core region (35°N–45°N, 30°E–120°E) (black box in Fig. 1) at 200 hPa, using the ordinary least squares method[39]. The Eurasian subtropical westerly jet is characterised by two centres over the West and East Asia (Fig. 1c)[4,5]. We analyzed the climatology and trends of both the West Asian jet (30°E–70°E) and East Asian jet (70°E–120°E) based on four reanalyses and results are shown in Supplementary Fig. S1. The vertical structures of the climatology, linear trends over two sectors and time series over two regions at 200 hPa show similar features. Therefore, in this study, we combined these two jets and defined the Eurasian subtropical westerly jet (ESWJ) index which covers both the West Asian and East Asian subtropical jets. Here we focused on data since 1979 (1980 for MERRA2) after the introduction of satellite data in reanalyses and used full length during the last four decades from 1979 (1980) to 2019 (41 years, 40 years for MERRA2). The significance of zonal wind trends was tested by the Mann–Kendall nonparametric method. The linear trend and 90% uncertainty range for the ESWJ index for each reanalysis were also estimated using the ordinary least squares method[52]. The mean trend of the ESWJ index was the arithmetic mean of trends based on four reanalyses.

**CMIP6 and DAMIP simulations**. We investigated the impacts of anthropogenic forcings on trends of westerlies and the summer ESWJ using multimodel simulations of the Coupled Model Intercomparison Project Phase 6 (CMIP6)[42], including both historical all forcing simulations (referred to as ALL: driven with changes in all anthropogenic and natural forcings) and single forcing simulations from the Detection and Attribution Model Intercomparison Project (DAMIP)[43]. Single forcing experiments include greenhouse gases (GHG) only (driven with changes in well-mixed greenhouse gas concentrations only), anthropogenic aerosol (AER) only (driven with changes in anthropogenic aerosol emissions), and natural

forcing (NAT) only (driven with changes in natural forcings only) simulations which were designed to estimate the contributions of different anthropogenic and natural forcings to observed global and regional climate changes. We selected eight models that have more than three members for all historical and single forcing simulations (Supplementary Table 1). They are the Beijing Climate Center Climate System Model (BCC-CSM2-MR)[53], the Canadian Earth System Model version 5 (CanESM5)[54], the sixth generation Centre National de Recherches Météorologique Coupled Model (CNRM-CM6-1)[55], the Goddard Insitute for Space Studies climate model (GISS-E2-1-G)[56], the Hadley Centre Global Environment Model version 3 (HadGEM3-GC31-LL)[57], the Institute Pierre-Simon Laplace Climate Model (IPSL-CM6A-LR)[58], the Model for Interdisciplinary Research on Climate version 6 (MIROC6)[59] and the Meteorological Research Institute Earth System Model (MRI-ESM2-0)[60].

We downloaded monthly mean variables from these simulations. Model simulations were interpolated to a common grid with a horizontal resolution of 1.875° longitude by 1.25° latitude before the analysis. We used monthly mean data to construct summer (June, July, August) means and analysed model-simulated trends of westerlies and the summer ESWJ index for the period 1979–2014 (36 years) in model simulations since historical all forcing simulations stopped at 2014. Trends in both reanalyses and model simulations were given by trends per decade to make it easy to compare them. The piControl simulations for these models have a total length of 5950 years with the forcing constant in time at 1850 values. They are used to assess the role of internal variability on ESWJ trends with 36-year-long non-overlapping segments (Supplementary Table 1).

We calculated trends for each member of model experiments with different forcings, constructed ensemble mean trends for different forcing experiments for each model, and then constructed the multimodel mean (MMM) by averaging eight model results (i.e. giving equal weight to each model) for different forcing simulations. The MMM trends in NAT simulations are based on seven model simulations since some variables in GISS-E2-1-G NAT simulations are not available in the database. The robustness of multimodel simulations was assessed if seven (six) out of eight (seven) models gave the same sign of trends in ALL, GHG, AER (NAT) simulations. We revealed that anthropogenic aerosols were likely the primary driver of the observed weakening trends of the summer ESWJ since 1979. We calculated trends of meridional temperature gradient (MTG: d$T$/d$y$, where $T$ is temperature and $y$ is the meridional distance) at pressure levels and at latitude-height cross-section to reveal the relationship between trends of westerlies and trends of MTG in the troposphere. The MMM mean MTG at the latitude-height cross-section in ALL simulations is based on seven model mean with CanESM5 excluded since CanESM5 shows large temperature trends over the Tibetan Plateau below 500 hPa and therefore shows inconsistent trends of MTG over the Tibetan Plateau in comparison with other models. The uncertainty for the multimodel mean ESWJ index trend is estimated based on $\sigma/\sqrt{N}$ where $\sigma$ is standard deviation of ESWJ index trends among different models and N is the number of models.

The trends of zonal wind at the latitude-height cross-section are also calculated from the trends of MTG by vertically integrating the thermal wind balance equation[34] $\frac{du}{dp} = \frac{R}{fp}\left(\frac{dT}{dy}\right)$, where $u$ is zonal mean zonal wind, $p$ is pressure, $R$ is the gas constant for dry air, $f$ is the Coriolis parameter, and d$T$/d$y$ is the MTG on constant pressure surfaces. A residual component is also computed from the difference between linear trends of the total zonal wind and zonal wind trends derived from the trends of MTG based on the thermal wind balance.

We introduced a meridional gradient index of downward surface solar radiation (SSR) and a meridional surface air temperature (SAT) gradient index to investigate the relationships between these two indices and the ESWJ index among multi-models and multimodel ensembles. The SSR gradient index is defined as the area-averaged difference between the region (20°–40°N and 0°–90°E) and region (40°–60°N and 0°–90°E) located to the south and north of the summer Eurasian climatological jet core (red and black boxes in Fig. 4a). SAT gradient index is defined as the difference between the region (20°–40°N and 30°–120°E) and region (40°–60°N and 30°–120°E) (red and black boxes in Fig. 4b) considering the downstream influence of aerosol emissions[61]. The gradient trends of these indices are the differences of trends between the south box and the north box.

The effective radiative forcing (ERF) of aerosols in Northern Hemisphere summer is taken from a recent study[62], which is available for six models of CanESM5, CNRM-CM6-1, GISS-E2-1-G, IPSL-CM6A-LR, MIROC6 and MRI-ESM2-0 and is used to investigate relationships between aerosol effective radiative forcing (ERF) and SSR gradient index trends.

## Data availability

ERA5 reanalysis is available at https://climate.copernicus.eu/climate-reanalysis. NCEP reanalysis is available at https://psl.noaa.gov/data/gridded/data.ncep.reanalysis.html. JRA55 reanalysis is available at https://jra.kishou.go.jp/JRA-55/index_en.html. MERRA2 reanalysis is available at https://gmao.gsfc.nasa.gov/reanalysis/MERRA-2/data_access/. CRUT4.05 data set is available at https://crudata.uea.ac.uk/cru/data/hrg/. The CMIP6 and DAMIP simulations analyzed in this study are versions archived at the Centre for Environmental Data Analysis (CEDA) and they are available at https://help.ceda.ac.uk/article/4801-cmip6-data.

## Code availability

All relevant codes used in this work are available, upon request, from the corresponding author B.D.

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

## Acknowledgements

This work was supported by the Natural Environment Research Council (NERC) North Atlantic Climate System Integrated Study (ACSIS) project. B.D., R.S., L.S. and B.H. are supported by the UK National Centre for Atmospheric Science, funded by the Natural Environment Research Council. We acknowledge the World Climate Research Programme, which, through its Working Group on Coupled Modelling, coordinated and promoted CMIP6.

## Author contributions

B.D. and R.T.S. designed research. B.D. carried out analysis. B.D., R.T.S., L.S. and B.H. worked together on the interpretation of the results and wrote the paper.

## Competing interests

The authors declare no competing interests.
