## [Peer Review File · Nature Communications]

REVIEWER COMMENTS

Reviewer #1 (Remarks to the Author):

This study focused on the decadal weakening of the summer Eurasian westerly jet. Authors addressed that the anthropogenic aerosols were likely the primary driver of the weakening ESWJ, due to the reduced meridional temperature gradient. The analysis and manuscript are broadly fine, but there are a couple of reasonably significant changes. A major revision is suggested before I make further recommendations. The detailed comments are listed.

Specific Comments:

1, The examination based on observation datasets.

Does the relationship between the aerosol emission and westerly jet exist in the observation datasets? I suggest authors should give the whole figure of the AOD. As previous studies indicated, there are several emission centers, as East Asia, north Africa and Europe. Audiences would also be interested in where is the sensitive region. As shown in Fig.5h, it supposes that East Asia, north Africa might play important role.

2, The quantitative analysis on the relative contribution.

It is quite difficult to understand the relative contribution of aerosol to the weakening of the ESWJ based on Fig.3. Quantitative analysis is needed to answer the question how many percentages of the trend in the ESWJ can be explained by the aerosol emission?

3, The possible mechanisms.

Authors investigate the possible mechanisms from the perspective of the MTG based on the thermal wind balance. The effect of the transport of eddy heat and momentum as well as the atmospheric baroclinicity are also important (Kuang et al. 2014). I suggest authors examine both the thermal and dynamic factors.

4, The model uncertainties.

Fig.4b, it seems that the SAT trends based on the ALL simulations are not similar as the ERA5 reanalysis. Authors should firstly examine the performance of the model.

Spatial patterns of the linear trends of surface air temperature in summer during 1979-2014 from the ERA5 reanalysis datasets

Fig.3, as the range of the trend for the AER and ALL simulations are large. The model uncertainties should be paid attention and author should address the credibility of the results.

5, Authors choose 500hPa MTG to understanding the possible mechanism of the weakening of EAWJ. Due to the thermal wind balance, I suggest authors show the value averaged vertically from surface to 200 hPa as Zhang et al. (2006) indicated.

6, Why authors choose the Eurasian sector (30°E-120°E)? As shown in Fig.1a, there are two large wind centers located in the Eurasian sector defined by authors. Combined with Fig.1c, the linear trend is significant for the eastern wind center. Are the results sensitive to a change in the chosen of the regions, e.g., separately analyzing the two wind centers?

7, Figure 4, right panels: the linear trends of precipitation for DAMIP simulations can be deleted, since they are not so relevant to understand the main conclusions.

8, As shown in Fig.1c, Fig.2, Fig.5, in the high latitudes around 60°N, it indicates an enhancing of summer East Asian polar front jet, which has been found in Xiao et al. (2020). In fact, recently, many studies have recognized the importance of the concurrent variation of the polar-front jet and subtropical jet (e.g., Liao and Zhang 2013; Huang et al. 2014; Li and Zhang 2014; Huang et al. 2017; Xue and Zhang 2017). I suggest authors can add some discussion on the response of the concurrent variation of the two jets to the aerosol emissions.

References:

Huang, D., J. Zhu, Y. Zhang, and A. Huang, 2014: The different configurations of the East Asian polar front jet and subtropical jet and the associated rainfall anomalies over eastern China in summer. *J. Clim.*, 27, 8205–8220, <https://doi.org/10.1175/JCLI-D-14-00067.1>.

—, and Coauthors, 2017: Recent Winter Precipitation Changes over Eastern China in Different Warming Periods and the Associated East Asian Jets and Oceanic Conditions. *J. Clim.*, 30, 4443–4462, <https://doi.org/10.1175/JCLI-D-16-0517.1>.

Kuang, X., Y. Zhang, Y. Huang, and D. Huang, 2014: Spatial differences in seasonal variation of the upper-tropospheric jet stream in the Northern Hemisphere and its thermal dynamic mechanism. *Theor. Appl. Climatol.*, 117, 103–112, <https://doi.org/10.1007/s00704-013-0994-x>.

Li, L., and Y. Zhang, 2014: Effects of different configurations of the East Asian subtropical and polar front jets on precipitation during Meiyu season. *J. Clim.*, <https://doi.org/10.1175/JCLI-D-14-00021.1>.

Liao, Z., and Y. Zhang, 2013: Concurrent variation between the East Asian subtropical jet and polar front jet during persistent snowstorm period in 2008 winter over southern China. *J. Geophys. Res. Atmos.*, 118, 6360–6373, <https://doi.org/10.1002/jgrd.50558>.

Xiao, X., D. Huang, B. Yang, J. Zhu, P. Yan, and Y. Zhang, 2020: Contributions of Different Combinations of the IPO and AMO to the Concurrent Variations of Summer East Asian Jets. *J. Clim.*, 33, 7967–7982, <https://doi.org/10.1175/jcli-d-19-0366.1>.

Xue, D., and Y. Zhang, 2017: Concurrent variations in the location and intensity of the Asian winter jet streams and the possible mechanism. *Clim. Dyn.*, 49, 37–52, <https://doi.org/10.1007/s00382-016-3325-y>.

Zhang, Y., X. Kuang, W. Guo, and T. Zhou, 2006: Seasonal evolution of the upper-tropospheric westerly jet core over East Asia. *Geophys. Res. Lett.*, 33, L11708, <https://doi.org/10.1029/2006GL026377>.

Reviewer #2 (Remarks to the Author):

Review of “Recent decadal weakening of the summer Eurasian westerly jet attributable to anthropogenic aerosol emissions” by Dong et al.

Summary

This paper presents a relatively straightforward analysis showing recent (1979-onwards) weakening of the Eurasian summer jet is likely due to anthropogenic aerosols. The paper is well written, clear and concise. The analysis is also thorough and convincing. It adds to the growing number of studies that show anthropogenic aerosols can perturb large-scale atmospheric circulation, including the jets.

Comments

Fig. 2 caption. “...indicate regions where at least 7 models showing the same sign of trends...” Maybe note that this is 7 out of 8 models?

The mechanism should probably be elaborated upon, based on prior studies that have used a similar argument. That is, changes in aerosols (which are spatially heterogeneous) drive heterogeneous warming/cooling, which impacts the tropospheric meridional temperature gradient, which in turn is linked to the upper levels winds/jets via thermal wind balance.

Papers that have previously used this mechanism to argue for aerosol-induced perturbations to jets include:

Allen, R. J., and Ajoku, O. (2016), Future aerosol reductions and widening of the northern tropical belt, *J. Geophys. Res. Atmos.*, 121, 6765–6786, doi:10.1002/2016JD024803.

Allen, R. J., Sherwood, S. C., Norris, J. R., and Zender, C. S.: The equilibrium response to idealized thermal forcings in a comprehensive GCM: implications for recent tropical expansion, *Atmos. Chem. Phys.*, 12, 4795–4816, <https://doi.org/10.5194/acp-12-4795-2012>, 2012.

Allen, R. J., Lamarque, J.-F., Watson-Parris, D., & Olivie, D. (2020). Assessing California wintertime precipitation responses to various climate drivers. *Journal of Geophysical Research: Atmospheres*, 125, e2019JD031736. <https://doi.org/10.1029/2019JD031736>

Possible model shortcomings, including for example aerosol forcing/emissions/climate impacts should also probably be discussed. For example:

CMIP6 AA emissions (from CEDS) likely underestimate the recent decrease in East Asia AA emissions (Wang et al., 2021).

Wang, Z., Lin, L., Xu, Y. et al. Incorrect Asian aerosols affecting the attribution and projection of regional climate change in CMIP6 models. *npj Clim Atmos Sci* 4, 2 (2021). <https://doi.org/10.1038/s41612-020-00159-2>

CMIP6 models may overestimate AA-driven cooling of the NH mid-latitudes during the mid-20th century (Zhang et al., 2021).

Smith and Forster (2021) show that the enhanced cooling from 1960-1990 in CMIP6 is due to both a marginally more negative AA forcing as well as a weaker positive GHG forcing. Furthermore, CMIP6 models possess a larger climate sensitivity than older versions, which amplifies these forcing differences—including more cooling from aerosols (Smith and Forster, 2021).

Zhang, J., Furtado, K., Turnock, S. T., Mulcahy, J. P., Wilcox, L. J., Booth, B. B., Sexton, D., Wu, T., Zhang, F., and Liu, Q.: The role of anthropogenic aerosols in the anomalous cooling from 1960 to

1990 in the CMIP6 Earth System Models, Atmos. Chem. Phys. Discuss. [preprint],
<https://doi.org/10.5194/acp-2021-570>, in review, 2021.

Smith and Forster, GRL, 2021. Suppressed late-20th Century warming in CMIP6 models explained by forcing and feedback. Please cite this article as doi: 10.1029/2021GL094948.

The authors greatly appreciate the very helpful comments and suggestions of the two reviewers. In the following we present our point-by-point responses to their comments. The reviewers' comments are in black and our responses are in blue. We have made appropriate changes in the revised manuscript, taking all comments and suggestions into account in the revision.

REVIEWER COMMENTS

Reviewer #1 (Remarks to the Author):

This study focused on the decadal weakening of the summer Eurasian westerly jet. Authors addressed that the anthropogenic aerosols were likely the primary driver of the weakening ESWJ, due to the reduced meridional temperature gradient. The analysis and manuscript are broadly fine, but there a couple of reasonably significant changes. A major revision is suggested before I make further recommendations. The detailed comments are listed.

We thank the reviewer for these general comments and suggestions on improving our submitted manuscript.

Specific Comments:

1, The examination based on observation datasets.

Does the relationship between the aerosol emission and westerly jet exist in the observation datasets? I suggest authors should give the whole figure of the AOD. As previous studies indicated, there are several emission centers, as East Asia, north Africa and Europe. Audiences would also be interested in where is the sensitive region. As shown in Fig.5h, it supposes that East Asia, north Africa might play important role.

We thank the reviewer for this suggestion. We have analyzed trends of aerosol optical depth (AOD) based on MERRA2 reanalysis and also DAMIP AER experiments. Results for summer are shown in Figure R1 (a, e). The reanalysis shows decreases of AOD over North America and Europe and increases over East Africa, Middle East, South Asia, and East Asia (Fig. R1a). Model results capture these main features with similar magnitudes although there are some regional differences. These changes in AOD and aerosol-cloud interactions resulting from aerosol emission changes lead to increases in downward surface solar radiation over Europe and decreases over North Africa, South Asia and East Asia (Fig. 4g in revision). Discussion of the changes in AOD has been added in lines 138-142 in the revised manuscript and Figure R1 is added as Supplementary Fig. S5.

Figure R1. It is added as Supplementary Fig. S5 in the revised manuscript. **The linear trends of aerosol optical depth (AOD) and surface air temperature (SAT) in summer (June, July, August) during the last four decades. a**, linear trends (decade^{-1}) of AOD based on MERRA2 reanalysis during 1980-2019. **b, c, d**, linear trends of SAT ($^{\circ}\text{C decade}^{-1}$) based on MERRA2 (1980-2019), ERA5 and CRUT4.05 (1979-2019) data sets. **e**, linear trends (decade^{-1}) of AOD based on DAMIP AER (7 models) simulations during 1979-2014. We show AOD in AER rather than ALL simulations because more AOD data is available for AER simulations. However AOD patterns are expected to be very similar in AER and ALL. **f**, linear trends of SAT ($^{\circ}\text{C decade}^{-1}$) based on CMIP6 (8 models) simulations during 1979-2014. Dots in **a, b, c, d** indicate regions where trends are statistically significant at the 10% level using the Mann-Kendall test. Dots in **e, f** indicate where at least 6 (7) out of 7 (8) models have the same sign of trend. See Methods for details of data sets and analysis.

2, The quantitative analysis on the relative contribution.

It is quite difficult to understand the relative contribution of aerosol to the weakening of the ESWJ based on Fig.3. Quantitative analysis is needed to answer the question how many percentages of the trend in the ESWJ can be explained by the aerosol emission?

In the manuscript, we stated that the multimodel mean (MMM) ESWJ index trend is $-0.34 \pm 0.07 \text{ m s}^{-1} \text{ decade}^{-1}$ in CMIP6 ALL forcing simulations. AER forcing induces a

weakening of the MMM ESWJ index with a value of $-0.36 \pm 0.04 \text{ m s}^{-1} \text{ decade}^{-1}$, whereas GHG and NAT forcing both induce a weak enhancement. Therefore, our best estimate is that AER forcing is responsible for 100% of the *forced* weakening of the ESWJ. There could also be a contribution from internal variability (which could explain the difference between forced trend and the observed trend of $-0.48 \text{ m s}^{-1} \text{ decade}^{-1}$ (mean value based on the four reanalysis data sets) but, as we discuss, there is only a 6.5% chance that internal variability alone could account for a weakening trend as large as that observed (see lines 114-120 in the revised manuscript).

3, The possible mechanisms.

Authors investigate the possible mechanisms from the perspective of the MTG based on the thermal wind balance. The effect of the transport of eddy heat and momentum as well as the atmospheric baroclinicity are also important (Kuang et al. 2014). I suggest authors examined both the thermal and dynamic factors.

We thank the reviewer for this suggestion. We have calculated the difference between linear trends of zonal wind (total change, right column of Fig.2) and zonal wind trends derived from thermal wind balance (right column of Fig.5). As shown in Figure R2, the magnitude of the residual is small in comparison with the thermal wind component. These results indicate that the trends of summer Eurasian subtropical westerly jet are primarily determined by changes in MTG in the lower and middle troposphere, being consistent with the study on the summer East Asian subtropical jet (e.g., Kuang et al. 2014). We have added some descriptions on this aspect in lines (184-189, 545-547) in the revised manuscript. Figure R2 is added as Supplementary Fig. S8 in revision.

Figure R2. As Supplementary Fig. S8 in the revised manuscript. **The multimodel mean (MMM) residuals of zonal wind trends at the latitude-height (hPa in pressure coordinate) cross section in summer (June, July, August).** Residuals of linear trends of zonal winds ($\text{m s}^{-1} \text{decade}^{-1}$), defined as the differences between linear trends of zonal wind (right panels of Fig. 2) and zonal winds derived from the cross-section MTG based on the thermal wind balance (right panels of Fig.5), at the latitude-height (hPa in pressure coordinate) cross section zonally averaged over the Eurasian sector (30°E - 120°E) with contours showing the corresponding climatology during 1979-2014. **a**, ALL simulations. **b**, GHG simulations. **c**, AER simulations. **d**, NAT simulations. See Methods for details of model simulations and analysis.

4, The model uncertainties.

Fig.4b, it seems that the SAT trends based on the ALL simulations are not similar as the ERA5 reanalysis. Authors should firstly examine the performance of the model.

Spatial patterns of the linear trends of surface air temperature in summer during 1979-2014 from the ERA5 reanalysis datasets

We have analyzed SAT trends in MERRA2, ERA5 reanalyses and SAT trends from CRU TS4.05 observational dataset. Results are shown in Figure R1. Observations and reanalyses show large warming trends over North Africa, southern Europe, Middle East and East Asia with weak trends over parts of North America and Central Asia. The multimodel mean (MMM) SAT trends in CMIP6 ALL forcing simulations show similar warming trends over North Africa, southern Europe, Middle East and East Asia, but warming trends over North America and Central Asia are overestimated. We have added a description of these comparisons in lines 148-152 in the revised manuscript and Figure R1 is added as Supplementary Figure S5.

Fig.3, as the range of the trend for the AER and ALL simulations are large. The model uncertainties should be paid attention and author should address the credibility of the results.

We agree that there is substantial spread amongst different models and ensemble members, reflecting both differing sensitivities to forcing and internal variability. Figure 3 provides quantification of both these contributions and shows that it is very unlikely (probability of 6.5%) that the observed weakening trend in the ESWJ could be explained by internal variability alone. Moreover, comparison of Figs 1 and 2 demonstrates that the models simulate a response to AER forcing which exhibits a three-dimensional spatial structure that is consistent with the observed changes in the ESWJ. This consistency provides compelling evidence to support our conclusion that changes in AER forcing were the primary driver of the weakening ESWJ. See also our response to point 2 above.

5, Authors choose 500hPa MTG to understanding the possible mechanism of the weakening of EAWJ. Due to the thermal wind balance, I suggest authors show the value averaged vertically from surface to 200 hPa as Zhang et al. (2006) indicated.

500 hPa MTG (left panels of Fig. 5) is used to show spatial structure of MTG in the middle troposphere. In the paper, we argue that it is the changes of MTG in the troposphere below 200 hPa that are responsible for zonal wind trends in the upper troposphere. These arguments are supported by the latitude-height cross section of zonally averaged MTG (middle panels of Fig. 5) and associated zonal wind changes through thermal wind balance (right panels of Fig. 5). Based on reviewer's suggestion, we have calculated vertically averaged MTG from surface to 200 hPa and results are given Figure R3. These show a reduced MTG in mid-latitudes over the Eurasian continent in ALL and AER simulations, being similar to those shown at 500 hPa (left panels in Fig.5) but with smaller magnitudes. However, we do not think this figure adds much extra information and therefore it has not been included in the revised manuscript.

Figure R3: The multimodel mean (MMM) linear trends of vertically averaged meridional temperature gradient (MTG) in summer (June, July, August) during 1979-2014 in CMIP6 (DAMIP) simulations. Linear trends of vertically averaged meridional temperature gradient (MTG) ($\text{K per } 1000 \text{ km decade}^{-1}$) with black boxes highlighting the region ($35^{\circ}\text{N}-45^{\circ}\text{N}$, $30^{\circ}\text{E}-120^{\circ}\text{E}$) where ESWJ index is calculated. **a**, ALL simulations. **b**, GHG simulations. **c**, AER simulations. **d**, NAT simulations. Only 7 models are included in panels **a**, **d**. Dots highlight regions where at least 7 (6) out of 8 (7) models showing the same sign of trends for GHG, and AER (ALL and NAT) simulations. See Methods for details of model simulations and analysis.

6, Why authors choose the Eurasian sector ($30^{\circ}\text{E}-120^{\circ}\text{E}$)? As shown in Fig.1a, there are two large wind centers located in the Eurasian sector defined by authors. Combined with Fig.1c, the linear trend is significant for the eastern wind center. Are the results sensitive to a change in the chosen of the regions, e.g., separately analyzing the two wind centers?

The Eurasian subtropical westerly jet is characterised by two centres over West and East Asia (Fig. 1c)⁴⁻⁵. We have further analyzed climatology and trends of both West Asian jet

(30°E-70°E) and East Asian jet (70°E-120°E) based on four reanalyses and results are shown in Figure R4. The vertical structures of the climatology, linear trends over two sectors and time series over two regions at 200 hPa show similar features. Therefore, in this study, we combined these two jets and defined the Eurasian subtropical westerly jet (ESWJ) index which covers both West Asian and East Asian subtropical jets. We have added these discussions in lines 483-489 and Figure R4 as Supplementary Fig. S1 in the revised manuscript.

Figure R4. Added as Supplementary Fig. S1 in the revised manuscript. **The linear trends of zonal wind in summer (June, July, August) during the last four decades. a, b,** climatology (m s^{-1}) (time mean over 1979-2019) at the latitude-height (hPa in pressure coordinate) cross section zonally averaged over the West Asian (30°E-70°E) and East Asian (70°E-120°E) sectors based on ERA5 reanalysis. **c, d,** linear trends ($\text{m s}^{-1} \text{ decade}^{-1}$) with contours showing climatology. **e, f,** time series of the West Asian subtropical and East Asian subtropical westerly indices, defined as the area averaged zonal wind over the regions of 35°N-45°N, 30°E-70°E, and 35°N-45°N, 70°E-120°E at 200 hPa, based on four reanalyses and corresponding linear trends during 1979-2019 (1980-2019 for MERRA2). Thick black lines in **c, d** indicate regions where trends are statistically significant at the 10% level using the Mann-Kendall test. See Methods for details of data sets and analysis.

7, Figure 4, right panels: the linear trends of precipitation for DAMIP simulations can be deleted, since they are not so relevant to understand the main conclusions.

We thank the reviewer for this suggestion. However, we prefer to keep precipitation trends since there are large contrasts in the tropical precipitations trends between ALL and AER simulations. Our results suggest that differences in the precipitations between the ALL and AER simulations do not play a major role in the weakening of the ESWJ (see lines 180-183).

8, As shown in Fig.1c, Fig.2, Fig.5, in the high latitudes around 60°N, it indicates an enhancing of summer East Asian polar front jet, which has been found in Xiao et al. (2020). In fact, recently, many studies have recognized the importance of the concurrent variation of the polar-front jet and subtropical jet (e.g., Liao and Zhang 2013; Huang et al. 2014; Li and Zhang 2014; Huang et al. 2017; Xue and Zhang 2017). I suggest authors can add some discussion on the response of the concurrent variation of the two jets to the aerosol emissions.

We thank the reviewer for this suggestion and recommended references. We have added some discussions on the subpolar jet in observations (see lines 79-81 in the revised manuscript) and model simulations (see line 90, lines 107-108 in the revised manuscript). We have cited Zhang et al. (2006), Kuang et al (2014), Huang et al (2014), Li and Zhang (2014), and Xiao et al. (2020) (see refs 1, 2, 12, 13, and 16 in the revised manuscript), but have not cited the other three since they are about jet changes in winter and their impacts on winter climate.

References:

- Huang, D., J. Zhu, Y. Zhang, and A. Huang, 2014: The different configurations of the East Asian polar front jet and subtropical jet and the associated rainfall anomalies over eastern China in summer. *J. Clim.*, 27, 8205–8220, <https://doi.org/10.1175/JCLI-D-14-00067.1>.
- , and Coauthors, 2017: Recent Winter Precipitation Changes over Eastern China in Different Warming Periods and the Associated East Asian Jets and Oceanic Conditions. *J. Clim.*, 30, 4443–4462, <https://doi.org/10.1175/JCLI-D-16-0517.1>.
- Kuang, X., Y. Zhang, Y. Huang, and D. Huang, 2014: Spatial differences in seasonal variation of the upper-tropospheric jet stream in the Northern Hemisphere and its thermal dynamic mechanism. *Theor. Appl. Climatol.*, 117, 103–112, <https://doi.org/10.1007/s00704-013-0994-x>.
- Li, L., and Y. Zhang, 2014: Effects of different configurations of the East Asian subtropical and polar front jets on precipitation during Meiyu season. *J. Clim.*, <https://doi.org/10.1175/JCLI-D-14-00021.1>.
- Liao, Z., and Y. Zhang, 2013: Concurrent variation between the East Asian subtropical jet and polar front jet during persistent snowstorm period in 2008 winter over southern China. *J. Geophys. Res. Atmos.*, 118, 6360–6373, <https://doi.org/10.1002/jgrd.50558>.
- Xiao, X., D. Huang, B. Yang, J. Zhu, P. Yan, and Y. Zhang, 2020: Contributions of Different Combinations of the IPO and AMO to the Concurrent Variations of Summer East Asian Jets. *J. Clim.*, 33, 7967–7982, <https://doi.org/10.1175/jcli-d-19-0366.1>.

Xue, D., and Y. Zhang, 2017: Concurrent variations in the location and intensity of the Asian winter jet streams and the possible mechanism. *Clim. Dyn.*, 49, 37–52, <https://doi.org/10.1007/s00382-016-3325-y>.

Zhang, Y., X. Kuang, W. Guo, and T. Zhou, 2006: Seasonal evolution of the upper-tropospheric westerly jet core over East Asia. *Geophys. Res. Lett.*, 33, L11708, <https://doi.org/10.1029/2006GL026377>.

Reviewer #2 (Remarks to the Author):

Review of “Recent decadal weakening of the summer Eurasian westerly jet attributable to anthropogenic aerosol emissions” by Dong et al.

Summary

This paper presents a relatively straightforward analysis showing recent (1979-onwards) weakening of the Eurasian summer jet is likely due to anthropogenic aerosols. The paper is well written, clear and concise. The analysis is also thorough and convincing. It adds to the growing number of studies that show anthropogenic aerosols can perturb large-scale atmospheric circulation, including the jets.

We thank the reviewer for a very good summary of our study and these positive remarks on our submitted manuscript.

Comments

Fig. 2 caption. “...indicate regions where at least 7 models showing the same sign of trends...” Maybe note that this is 7 out of 8 models?

Agree and we have made changes in all figure captions.

The mechanism should probably be elaborated upon, based on prior studies that have used a similar argument. That is, changes in aerosols (which are spatially heterogeneous) drive heterogeneous warming/cooling, which impacts the tropospheric meridional temperature gradient, which in turn is linked to the upper levels winds/jets via thermal wind balance.

Thank you for this summary on the physical mechanism involved. We agree with your suggestions that more previous studies should be cited.

Papers that have previously used this mechanism to argue for aerosol-induced perturbations to jets include:

Allen, R. J., and Ajoku, O. (2016), Future aerosol reductions and widening of the northern tropical belt, *J. Geophys. Res. Atmos.*, 121, 6765– 6786, doi:10.1002/2016JD024803.

Allen, R. J., Sherwood, S. C., Norris, J. R., and Zender, C. S.: The equilibrium response to idealized thermal forcings in a comprehensive GCM: implications for recent tropical expansion, *Atmos. Chem. Phys.*, 12, 4795–4816, <https://doi.org/10.5194/acp-12-4795-2012>, 2012.

Allen, R. J., Lamarque, J.-F., Watson-Parris, D., & Olivie, D. (2020). Assessing California wintertime precipitation responses to various climate drivers. *Journal of Geophysical Research: Atmospheres*, 125, e2019JD031736. <https://doi.org/10.1029/2019JD031736>

We thank the reviewer for this comment and recommended references. They are all cited in the revised manuscript (references 27, 28, and 29).

Possible model shortcomings, including for example aerosol forcing/emissions/climate impacts should also probably be discussed. For example:

CMIP6 AA emissions (from CEDS) likely underestimate the recent decrease in East Asia AA emissions (Wang et al., 2021).

Wang, Z., Lin, L., Xu, Y. et al. Incorrect Asian aerosols affecting the attribution and projection of regional climate change in CMIP6 models. *npj Clim Atmos Sci* 4, 2 (2021). <https://doi.org/10.1038/s41612-020-00159-2>

CMIP6 models may overestimate AA-driven cooling of the NH mid-latitudes during the mid-20th century (Zhang et al., 2021).

Smith and Forster (2021) show that the enhanced cooling from 1960-1990 in CMIP6 is due to both a marginally more negative AA forcing as well as a weaker positive GHG forcing. Furthermore, CMIP6 models possess a larger climate sensitivity than older versions, which amplifies these forcing differences—including more cooling from aerosols (Smith and Forster, 2021).

Zhang, J., Furtado, K., Turnock, S. T., Mulcahy, J. P., Wilcox, L. J., Booth, B. B., Sexton, D., Wu, T., Zhang, F., and Liu, Q.: The role of anthropogenic aerosols in the anomalous cooling from 1960 to 1990 in the CMIP6 Earth System Models, *Atmos. Chem. Phys. Discuss.* [preprint], <https://doi.org/10.5194/acp-2021-570>, in review, 2021.

Smith and Forster, *GRL*, 2021. Suppressed late-20th Century warming in CMIP6 models explained by forcing and feedback. Please cite this article as doi: 10.1029/2021GL094948.

We thank the reviewer for this comment and recommended references. We acknowledge that aerosol radiative forcing and climate sensitivity might be too strong in some of the

CMIP6 models and, if so, the simulated ESWJ response to this forcing might also be somewhat stronger in the models than in the real world. However, our results show that the observed weakening (i) cannot be explained as a response to GHG or NAT forcing, (ii) is consistent in spatial structure with the simulated response to AER forcing, and (iii) is unlikely to be explained by internal variability alone. This suggests that any errors in the model forcing or sensitivity might influence our assessment of the quantitative contribution of AER forcing to changes in the ESWJ but would be unlikely to change our key conclusion that changes in anthropogenic aerosol pre-cursor emissions were the primary driver of weakening of the summer ESWJ over the last four decades. We have added a paragraph to discuss these issues (see lines 245-255) and cited the recommended references (references 46, 47, 50) in the revised manuscript. We also added the following two references (references 48, 49) in revision.

Forster, P. *et al.* Evaluating adjusted forcing and model spread for historical and future scenarios in the CMIP5 generation of climate models. *J. Geophys. Res. Atmos.* **118**, 1139–1150 (2013). <https://doi.org/10.1002/jgrd.50174>

Flynn, C. M., & and Mauritsen, T. On the climate sensitivity and historical warming evolution in recent coupled model ensembles, *Atmos. Chem. Phys.* **20**, 7829-7842, (2020). <https://doi.org/10.5194/acp-2019-1175>

REVIEWERS' COMMENTS

Reviewer #1 (Remarks to the Author):

Authors have revised the manuscript carefully and the responses to the reviewer's comments are reasonable. I don't have any other comments. The manuscript is suitable for publication in the journal.

Reviewer #2 (Remarks to the Author):

This is my second review of the manuscript. The authors have satisfactorily addressed my prior comments/concerns. I recommend that the paper be accepted.